# Interaction between the Participation in and the Impact on Mental Health Service Users and Their Relatives of a Multicomponent Empowerment-Based Psychosocial Intervention

**DOI:** 10.3390/ijerph192113935

**Published:** 2022-10-26

**Authors:** Francisco José Eiroa-Orosa, Maria Jesús San Pío, Gemma Marcet, Isabela Sibuet, Emilio Rojo

**Affiliations:** 1First-Person Research Group, Veus, Catalan Federation of 1st Person Mental Health Organisations, 08035 Barcelona, Catalonia, Spain; 2Section of Personality, Assessment and Psychological Treatment, Department of Clinical Psychology and Psychobiology, Faculty of Psychology, University of Barcelona, 08035 Barcelona, Catalonia, Spain; 3Federation Mental Health Catalonia, 08002 Barcelona, Catalonia, Spain; 4Hospital Benito Menni CASM, Sisters Hospitallers, 08830 Sant Boi de Llobregat, Catalonia, Spain; 5Department of Psychiatry, International University of Catalonia, 08195 Sant Cugat del Vallès, Catalonia, Spain

**Keywords:** community care, integrated care, mental health, service users and relatives’ interventions, service users and relatives’ interaction

## Abstract

Relatives play an important role in the recovery journey of mental health service users. Interventions directed either at service users or their relatives may influence the other person as well. The project ‘Activa’t per la salut mental’ (Get active for mental health) consisted of a series of four interventions addressed at people diagnosed with mental disorders and their relatives to help them in their recovery process, increasing their agency and quality of life. The main objective of the present study is to evaluate the interaction of the participation of service users on their relatives’ outcomes and vice versa. The impact of the project was evaluated within a randomised controlled trial. The treatment group had access to all the circuit interventions, while the control group received treatment as usual and could only access one of the interventions. All participants were evaluated at baseline, six months, and twelve months after the end of the first intervention. Service users were evaluated with the Stages of Recovery Instrument, and relatives with the Family Burden Interview Schedule II and the Duke-UNC-11 questionnaires. The interaction of participation and impact between service users and their relatives was analysed by means of correlational analyses within the intervention group (*n* = 111, service users mean age = 40.6, 40% women; relatives mean age = 56.7, 72% women). Service users’ baseline characteristics (being in a relationship, educational level, employment, and younger age) influenced in the level of participation of relatives and vice versa (lower educational level). The results also indicated correlations between participation and outcomes at various points as well as the evolution of service users’ recovery and the care burden of relatives. Service users’ participation levels interacted with the decrease of relatives’ frequency of burden and the first steps of their own recovery journey (moratorium, awareness, and preparation) while relative’s participation just interacted with the evolution of two stages of service users’ recovery levels (preparation and growth). These results can be extremely helpful in fostering interactive benefits in future projects addressing the wellbeing of mental health service users and their relatives. Future studies could use specific designs to explore the directionality of the causality of these effects.

## 1. Introduction

In recent decades, there has been a shift towards community-focused systems of mental health care [1]. For instance, the Recovery movement, which started in the early 1990s, has been highly influential in the recognition of the social dimensions of psychological distress and the redirection of interventions beyond targeting just symptomatology [2]. Recovery was defined as a personal, unique, and multidimensional process of changing one’s attitudes, values, feelings, goals, abilities, and/or roles leading to living a satisfying and hopeful life, despite the potential limitations caused by mental disorders [3]. Interventions conceived within this movement are directed towards raising awareness among service users and their communities to become active agents of recovery.

However, this shift has occurred without ensuring that service users’ relatives had the adequate structures and instruments to be able to be helpful in the recovery process, maintaining their own health and wellbeing as well [4,5]. Relatives play an important role in the recovery process by providing moral and practical support as well as motivation [6,7]. If not properly addressed, the unexpected responsibility for the family can have deleterious consequences for both service users and their relatives [8,9,10]. Therefore, and within the progressive focus on community, it has become evident that service users in community settings as well as their relatives need specific interventions [11]. It can also be hypothesized that if one of the members engaged in an intervention, the other member could benefit as well from their improvements.

The fact that targeting relatives can improve the progression of service users is a well-known phenomenon. Studies describing interventions directed to relatives measuring service users’ outcomes show clear improvements [12]. Relatedly, there are some precedents of research on interventions designed to offer psychoeducation to service users and their relatives as a unit [13,14,15]. However, most reported data focus on just one of the parts [12,14,16]. Results seem to indicate positive impacts for relatives in terms of morbidities, burden, and negative experiences [14]. Nevertheless, when taken together, the impact seems to be clearer for service users’ clinical status than for relatives’ burden [13].

### The Present Study

To fill the gap in recovery care directed towards families with a member diagnosed with a mental disorder, an integrated multicomponent intervention was designed by the Catalan Health Department together with local federations of organisations managed by service users and their relatives. The intervention was aimed at helping in the recovery process, increasing the agency and quality of life of both service users diagnosed with mental health disorders and their relatives. The project was named ‘Activa’t per la Salut Mental’ (Get Active for Mental Health in English; referred to as Activa’t from now on). It consists of a succession of four interventions targeted to mental health service users and their relatives.

The implementation of this multicomponent intervention was evaluated in terms of the efficacy of the circuit on participating users and one of their relatives. It became clear that the intervention is effective in reducing some aspects of relatives’ burden in relatives and in fostering recovery in service users, although long-term differences between control and intervention groups did not show statistical significance [17]. However, the interaction of the participation and impact of the interventions between the two members of the participating families remains unexplored. Therefore, the present study aims at establishing the mutual interaction effects of the participation and outcomes of service users and their relatives benefiting from this multicomponent intervention. We hypothesized that the progress in the recovery path of service users influenced on the reduction of the burden perceived by their relatives and that both phenomena was associated with the level of participation in the activities of the circuit.

## 2. Materials and Methods

### 2.1. Trial Design

The present study is a correlational longitudinal study based on the data collected on the Activa’t implementation Randomised Controlled Trial (RCT). The random allocation of the implementation study was 111 family units (1 service user and 1 relative, see below) to both the control group and the treatment groups (total *n* = 222 family units). For the current analyses, we focused on participants that received the full Activa’t intervention, i.e., the 111 family units allocated to the treatment group. The Clinical Research Ethics committees of all participating mental health centres approved the study. For more details, please consult the ISRCTN registry number 15181312 (https://doi.org/10.1186/ISRCTN15181312).

### 2.2. Participants

Participants were recruited from 12 mental health centres from different territories in Catalonia. Participation was organized through family units. A family unit consisted of one mental health service user and a relative who was involved in his or her care. Mental health centres selected potential participants from their databases, following selection criteria. Potential participants were informed about the study either by phone, individually or through informative sessions organized by professionals from the project team together with mental health services.

Service users’ inclusion criteria were: (a) being diagnosed as per ICD-10 [18] criteria with schizophrenia (F20), schizotypal (F21), delusional (F22), induced delusional (F24), schizoaffective (F25), bipolar (F31), or recurrent depressive (F33) disorders; (b) having had symptomatology for at least 2 years (without this implying necessarily the existence of a diagnosis or a treatment during the whole 2 years); (c) being between 18 and 63 years old; and (d) having a moderate or severe disability degree lower than 60 according to the Global Assessment of Functioning scale (Axis V of the Diagnostic and Statistical Manual of Mental Disorders, Fourth Edition Text Revision criteria [19]). Exclusion criteria for service users were: (a) diagnosis of a borderline personality disorder; (b) comorbidity with intellectual disability; (c) presence of severe associated somatic pathology; (d) being incapacitated and having tutelage by a protection institution, or being resident of nursing homes, protected housing, or long stay units, (e) having received structured psychoeducation or having participated in mutual support groups in the last 12 months. Relatives had to fulfil the following inclusion criteria: (a) being the main carer of a person with the mental disorder diagnoses specified above, and (b) having the practical availability to participate in all the Activa’t activities. The exclusion criteria for relatives were: (a) suffering from a psychiatric or severe somatic non-compensated disorder (including moderate and severe alcohol dependence requiring intervention), and (b) having received structured psychoeducation in the last 12 months.

All participants (service users and relatives separately) provided informed consent and were asked to fill demographic and baseline outcome questionnaires prior to randomisation. All participants filled outcome questionnaires again at six and twelve months after the treatment group completed the first block of interventions.

### 2.3. Sample Size

The sample size calculation was performed before the start of the recruitment phase, and it considered the maximum number of relatives that the mental health centres could treat (both intervention treatment and treatment as usual). The power size calculation indicated that 240 family units would allow for the detection of a Minimum Detectable Effect (MDE) of 0.36. The final sample size was 222 family units. As commented above, just the intervention group is used for the current study.

### 2.4. Interventions

The Activa’t programme is composed of four components, each of them led by different stakeholders. The first one was called Espai Situa’t (‘locate yourself’ space), a non-therapeutic counselling service provided by Salut Mental Catalunya (the Catalan federation of relatives and mental health service users). It offers response to any information demand about mental health that people may have and about the services available in each geographic area. The service is accessible to the public in general and could be used (or not) at any time of the trial by both the control and intervention groups. Twelve Espais Situa’t were created in community centres at all territories involved. The second component, which could be considered the first active intervention of the circuit, was a psychoeducation programme validated specifically for service users (Klave de Re [20,21]) or relatives (Training and education program for families and caregivers of people with serious mental disorders, PROENFA [22] by its acronym in Spanish), which were carried out in mental health centres by their clinical staff. The third component, considered the second step of the circuit, was an empowerment training programme called PROSPECT [23,24] provided also by Salut Mental Catalunya. The training aims at enhancing empowerment amongst relatives and users, and at developing empowerment-promoting skills among professionals, through structured programmes offered by previously trained peers. Finally, the fourth component consists of self- managed peer support groups linked to mental health advocacy groups. Peer-support groups were led by service users or by relatives (specific for each group) who had previously received specific training as group facilitators within the programme.

### 2.5. Outcomes

#### 2.5.1. Recovery Perceived by Mental Health Service Users

The Spanish version of the Stages of Recovery Instrument (STORI; [25]) was used to evaluate recovery among service users. STORI has 50 items and allows for the quantification of five consecutive stages of recovery. The stages are moratorium (stage of personal withdrawal characterised by a feeling of profound loss and absence of hope), awareness (incipient personal consciousness about how not everything is lost and perception of potential improvement), preparation (stage in which the person is conscious about the advantages and drawbacks of recovery and starts to think, in a practical level, about how to recover), rebuilding (stage in which the person works actively in their recovery, stating attainable objectives and regaining control over their own life) and growth (stage in which one lives a fruitful life, characterised by the personal regulation of the disorder, resilience and positive feelings about oneself). The items are quantified using a 6-point (0–5) Likert scale. We averaged the items of each stage, so scores range from 0 to 5. A high score on the items of a particular stage means that the person shows traits of that stage. From this follows that high scores on initial stages of the process, especially in moratorium, imply that the person is in an incipient moment of their recovery process, while high scores in later stages, such as rebuilding or growth, and imply that the person is in an advanced stage of their recovery. Reliabilities in our study ranged α = 0.734 to 0.897 at baseline, α = 0.693 to 0.900 at 6 months, and α = 0.717 to 0.924 at 12 months and were considered adequate.

#### 2.5.2. Perception of Social Support Perceived by Relatives

Social support was measured using the Spanish carer version [26] of the Duke-UNC-11 [27]. The instrument evaluates social support in three ways: confidential support, i.e., the degree to which the person has close people to communicate with; affective support, i.e., the intensity with which the person receives demonstrations of love, affect, and empathy; and global support, a composite score of the above. High scores on the Duke questionnaire represent a high perception of social support. The questionnaire is composed by 11 items, with scores ranging from 1 to 5 in a Likert scale. We averaged the items of each dimension so scores range 1–5. Reliabilities in our study ranged α = 0.718 to 0.873 at baseline, α = 0.796 to 0.904 at 6 months, and α = 0.828 to 0.915 at 12 months and were considered adequate.

#### 2.5.3. Care Burden Perceived by Relatives

The Spanish version [28] of the second version of the Family Burden Interview Schedule (FBIS-II) questionnaire [29] was used to assess the burden of care among participating relatives. The instrument assesses different dimensions of the burden of care: (A) service user’s routine, (B) service user’s disruptive behaviours, (C) financial burden, (D) caregiver’s routine, (E) concern, (F) available help, (G) repercussions on health, and (H) assessment of general burden. This information is assessed measuring presence of each burden element (yes/no), and just if the burden element is present, frequency (0–4), level of concern (0–3, just for items from modules A and B) and time (1–7, just for items from module A and overall in module B). Additionally, module C measures financial burden in euros. For each module in which it was applicable, the total presence of different types of burden, the sum and average scores of the frequency of burden, the sum and average scores of the concern experienced by the carer for each module, and a total sum of the money invested in the service user were calculated. This use of the questionnaire led to the creation of five frequency scores (help in daily-life activities, disruptive behaviours, change in carer’s routine, concern, and global burden), one cost score (Economic burden), and two concern scores (Concern about help in daily-life activities and Concern about disruptive behaviours). To facilitate understanding, total scores were created summing or averaging items across modules: a score with the total count of presence (yes/no) items that add burden to the relative across all modules (total presence of burden), scores of the sum and average frequency of burden, scores of the sum and average concern experienced by the carer and a sum of the time invested in the care of the service user. Additionally, using the latter, a total score was created multiplying the presence of burden by the averages of the frequency and concern and by the sum of time invested. High scores in all these indicators represent high burden of the carer. Reliabilities could only be calculated for scores that were replied by all participants using the same scale, i.e., just the presence (yes/no) items. Scores were acceptable at all time points (α = 0.659 at baseline, α = 0.634 at 6 months, and α = 0.628 at 12 months).

### 2.6. Analyses

To determine effects of higher or lower participation, we used participation data of intervention components 2 to 4 measured as total sessions attended. To facilitate analyses with sociodemographic data, we considered that participation could be divided into low participation (subjects below the median participation score) and high participation (subjects above the median participation score). Low and high participation groups baseline data were compared using Student’s *t* tests for continuous variables or chi-square and odds ratio for categorical variables. Subsequently, the differential outcome variables of the questionnaires from baseline to both follow-up points (i.e., t0–t6 and t0–t12) were correlated. Additionally, participation scores were correlated with all outcomes. Finally, all outcome variables were analysed as dependent variables using mixed model analyses with time (baseline, 6 months, 12 months) as within component and own participation and participation of the other member of the family unit as covariates. All analyses were performed with IBM SPSS 23.0. Significance was set at *p* < 0.05.

## 3. Results

A total of 222 participants (111 service users and 111 relatives) were randomised to the intervention group. Service users’ mean age was 40.6 years of age, 40% were women, 57.7% of them were diagnosed with schizophrenia, and 18.9% with bipolar disorder. Relatives mean age was 56.7 years of age, 72% of them were women.

### 3.1. Interaction of Participation with Participants’ Characteristics

The participation of service users was 19 ± 11.54 sessions (range 0–44, median 20) while their relatives’ was 15 ± 10.18 (range 0–36, median 15), with a mild correlation between both (r = 0.356, *p* < 0.0001). To show a clearer picture, Table 1 and Table 2 show characteristics of service users and relatives based on the dichotomised version of their total participation at the end of the programme’s implementation. As it can be seen, in the case of service users, educational level (at least secondary) was the only predictor of participation. In the case of relatives, no characteristic could predict the level of participation.

We then compared service users’ baseline parameters with relatives’ levels of participation and vice versa. Statistically significant results for relatives’ participation included service users’ relationship status (not being in a relationship), higher educational level, employment, and younger age. Relatives’ lower education level had a statistically significant influence on service users’ participation.

### 3.2. Correlational between Differential Outcome Scores

Results of the correlational analyses carried out with relatives’ by service users’ differential scores can be seen in Table 3. Moderate statistically significant results can be seen for DUKE’s global and confident social support scores with STORI’s moratorium and growth. FBIS-II Sum frequency of burden correlated negatively with STORI’s Rebuilding, Sum concern about burden positively with STORI’s Moratorium and negatively all the other STORI subscales, Total time negatively with Preparation, Rebuilding, and Growth, and the total score negatively with Moratorium and positively with all the other STORI subscales.

### 3.3. Interaction of Participation with Outcome Scores

Pearson correlations between service users’ and relatives’ participation and scores at all time points can be seen in Table 4. Participation of service users had a statistically significant correlation with various STORI scores at t6 and t12, one Duke score at t12 and various FBIS-II scores at t0 and, especially, t6. Participation of relatives correlated just with STORI’s Growth at t0 and t12. Correlations of participation by differential scores showed statistical signification just for STORI’s Moratorium t0–t6 (*r* = 0.233, *p* < 0.005) in the case of relatives.

We used mixed models to include the three time points of all outcome variables as single dependent variables and to include level of participation of service users or relatives as a covariate. Results can be seen in Table 5. Statistically significant interactions with levels of participation were found for service users’ with STORI’s Moratorium, Awareness, and Preparation as well as FBIS II’s sum and average frequency of burden. Relatives’ levels of participation interacted just with the evolution of STORI’s Preparation and Growth.

## 4. Discussion

This study aimed mainly at determining whether the participation of service users and relatives in the Activa’t programme interacted on each other’s outcome evolutions and if their evolutions correlated with each other. The results give insight into how an intervention for mental health service users can contribute to reducing the relatives’ burden and how an intervention for relatives can contribute to foster service users’ recovery. The optimization of this type of interventions can improve the prospects of mental health service users and their relatives.

We found that the level of participation of both service users and relatives correlated with baseline characteristics and scores, as well as with scores during the trial. Furthermore, differential scores of both groups correlated. Especially the evolution of more advanced stages of recovery achieved by service users correlated negatively with the evolution of the relatives’ total burden level. Finally, the evolution of the frequency of burden was found to interact with service users’ participation levels while the evolution of two stages of recovery interacted with relatives’ participation.

Relatives’ level of participation was influenced by several service users’ characteristics while service users’ participation levels were only influenced by relatives’ educational levels. Service users’ higher education levels positively influenced their participation and influenced that of their relatives, whereas relatives’ lower educational level influenced positively on service users’ participation levels. It is important to analyse these differences in the context of a country in which there have been great differences in access to education between close generations as changes in the system were carried out quickly especially during the late 1970s and early 1980s. While education was a luxury that few had access to in the case of most relatives according to their mean age; access, even to higher education, for most service users has been virtually universal. In addition, it should be considered that the elements of some interventions have been adapted to these generational differences. For example, while PROENFA’s materials are mostly audiovisual, those of Klave de Re include lengthy texts. The latter should be reconsidered to encourage the participation of service users with lower educational levels. The rest of the service users’ characteristics that influenced the participation of relatives show us that while the service user was more likely to not have a partner in the high-participation group, they were also more likely to be employed and to be younger. This may indicate that the involvement of relatives caring for older service users, with lower educational levels, being in a relationship and without occupation, needs additional stimulation since they are probably people who feel that the interventions do not bring them and their relative something significant.

The correlation between users’ participation and relatives’ participation was statistically significant. This positive correlation implies that the more one of the members of the family unit participated, the more the other member participated as well. However, the moderate effect size indicates that there were also some family units where this was not the case, and there was a high participation of relatives while low participation of service users and vice versa.

Correlational analyses with differential scores show a joint evolution of the progress in service users’ recovery path of and relatives’ reduction of burden. These correlations indicate in some scores, especially in the total score, that groups’ presence, concern, and frequency had the peculiarity of being more constant and stronger when the reduction of the load at 6 months was correlated with the increase in the stages of recovery at 12 months. These results indicate that the reduction in burden perception at six months occurred especially in family units where people were in a recovery process, but this did not manifest itself visibly until twelve months. Although the study design did not allow us to draw strong conclusions about the directionality of this relationship, we could hypothesize that if the burden elements softened at six months, this allowed a boost in the stages of recovery. In other words, first, more specific and identifiable elements of progress related to activities of daily living and disruptive behaviour are given, and this gives way to service users to be able to consider identity elements and possible life projects.

Service users’ participation levels correlated with various recovery scores at t6 and t12 as well as relatives’ Confident social support at t12 and various burden measures at t0 and t6 while relatives’ participation correlated just with Growth recovery stadium at t0 and t6. A look at the interaction of participation with the evolution of these scores shows us a similar picture. Service users’ participation covariates with their own early-stage recovery outcomes as well as with the perceived frequency of burden perceived by their relatives, while the latter’s participation covariates with the former’s Preparation and Growth recovery outcomes. Therefore, it seems that the participation of service users had a greater impact on the outcomes of their relatives, than vice versa. This clearly tells us that the involvement of service users was important both for their own recovery and to ease the burden on their relatives. Additionally, the participation of relatives is important for advancing the recovery of users, but not for the perception of their own burden. Perhaps this is because the program is rather focused on the recovery of service users. It would be important to add additional components of self-care in the psychoeducational contents for relatives that can help them to also have their own recovery path, lightening their perception of burden.

Although carried out with different designs, philosophies and focuses on other outcomes, these results somehow match the literature on the use of family psychoeducation where the effects were clearer on service users’ clinical status and disability than the burden of their relatives [13], and there are no clear predictors of burden relief [30] (either characteristics of the relative or the service user). Since our work is focused on the cross-interaction between participation and evolution, the impact of user participation on their own results and those of their relatives compared to the null influence of relatives’ participation on their results, could give ideas as to why, in general, the results of family psychoeducation are clearer in terms of service users’ outcomes than relatives’ ones. In any case, results on family psychoeducation and similar interventions are mixed, and there is literature that takes for granted the efficacy of these interventions on burden and related outcomes regardless of its effect on service users [12,16].

The present study has, of course, some limitations. The main one is that, although the original study was a randomised controlled trial, this was a correlational study carried with the active group of the latter. Furthermore, by working with participation data, we could be over representing the trends in groups whose participation was more intense. In terms of generalizability, the intervention is directed towards all the population who suffers or has relatives who suffer from a mental disorder. However, it is difficult to determine if the sample is representative of the general population that may need access to this intervention. The strict criteria for enrolment and demanding schedule of activities may have had a dissuasive effect for some individuals. However, the recruitment process included different territories, from both rural and urban areas, and the project was offered massively to most of the patients who fulfilled criteria.

## 5. Conclusions

The Activa’t interventions are effective in improving recovery in users and reducing some aspects of burden in relatives [17], especially when there is an active participation of the family unit in the development of the interventions. Thus, knowing that the participation of relatives fosters the participation of service users and their journey into recovery, the Activa’t components can be understood as key resources for joint recovery, understood as an advance in the stages of recovery of service users but also as a relief of the burden of relatives. It would be interesting to explore how this co-evolution occurs with a more specific design with the intention of increasing both the participation and the impact obtained by service users and their relatives in the future.

## Figures and Tables

**Table 1 ijerph-19-13935-t001:** Sociodemographic characteristics of service users recruited into the study by level of participation of both groups.

Service Users’ Baseline Characteristics	Service Users’ Participation							Relatives’ Participation						
	High (58)		Low (53)		Statistical Significance			High (58)		Low (53)		Statistical Significance		
	N	%	N	%	OR. 95% CI	*p*		N	%	N	%	OR. 95% CI	*p*	
Gender (% females)	22	37.9	22	41.5	0.86, 0.40–1.84	0.700		21	37.5	23	41.8	0.84, 0.39–1.79	0.642	
Couple (% in a relationship)	14	27.5	17	38.6	0.60, 0.25–1.43	0.246		10	20	21	46.7	0.29, 0.12–0.71	0.006 **	
Cohabitation (% autonomous)	15	32.6	18	38.3	1.28, 0.55–3.01	0.566		17	34	16	37.2	1.15, 0.49–2.7	0.747	
Education (% at least secondary)	37	63.8	21	39.6	2.69, 1.25–5.79	0.011 *		36	64.3	22	40	2.7, 1.25–5.8	0.010 **	
Employment situation (% employed)	3	5.2	1	1.9	2.84, 0.29–28.14	0.354		4	7.1	0	0	1.08, 1–1.16	0.044 *	
	**M**	**SD**	**M**	**SD**	**t**	* **p** *	**d**	**M**	**SD**	**M**	**SD**	**t**	* **p** *	**d**
Age (M ± SD)	39.38	10.10	42.02	10.85	−1.328	0.187	−0.25	36.77	10.44	44.58	9.07	−4.208	<0.0001 ***	−0.80

* *p* < 0.05, ** *p* < 0.01, *** *p* < 0.001.

**Table 2 ijerph-19-13935-t002:** Sociodemographic characteristics of relatives recruited into the study by level of participation of both groups.

Relative’s Baseline Characteristics	Relatives’ Participation							Service Users’ Participation						
	High (58)		Low (53)		Statistical Significance			High (58)		Low (53)		Statistical Significance		
	N	%	N	%	OR. 95% CI	*p*		N	%	N	%	OR. 95% CI	*p*	
Gender (% females)	43	76.8	37	67.3	1.61, 0.70–3.72	0.264		39	73.6	41	70.7	0.87, 0.0.38–1.99	0.734	
Couple (% in a relationship)	31	68.9	33	76.7	0.67, 0.26–1.73	0.408		28	70	36	75	1.29, 0.50–3.29	0.600	
Cohabitation (% autonomous)	25	53.2	14	40.0	0.59, 0.24–1.42	0.237		15	39.5	24	54.5	0.54, 0.23–1.31	0.173	
Education (% at least secondary)	33	58.9	28	50.9	1.38, 0.65–2.93	0.396		23	43.4	38	65.5	2.48, 1.15–5.34	0.019 *	
Employment situation (% employed)	23	41.1	23	41.8	0.97, 0.46–2.06	0.936		18	34	28	48.3	1.82, 0.84–3.91	0.126	
	**M**	**SD**	**M**	**SD**	**t**	* **p** *	**d**	**M**	**SD**	**M**	**SD**	**t**	* **p** *	**d**
Age (M ± SD)	57.96	9.91	55.36	15.06	1.077	0.284	0.20	56.71	11.30	56.64	14.24	0.027	0.979	0.01

* *p* < 0.05.

**Table 3 ijerph-19-13935-t003:** Pearson correlations between service users’ and relatives’ differential scores.

		Service User’s Outcomes (STORI Subscales)									
		Moratorium		Awareness		Preparation		Rebuilding		Growth	
Relatives’ Outcomes		t0–t6	t0–t12	t0–t6	t0–t12	t0–t6	t0–t12	t0–t6	t0–t12	t0–t6	t0–t12
**Duke**											
Affective Social Support	t0–t6	0.010	0.132	0.125	0.073	0.012	−0.051	0.030	0.008	0.053	−0.001
	t0–t12	0.063	0.245	−0.023	−0.025	−0.101	−0.067	−0.135	−0.055	−0.081	−0.092
Confident Social Support	t0–t6	0.038	0.254 *	−0.026	−0.098	−0.051	−0.229	−0.140	−0.207	−0.130	−0.236
	t0–t12	0.152	0.291 *	−0.105	−0.066	−0.091	−0.142	−0.194	−0.124	−0.261 *	−0.143
GLOBAL SOCIAL SUPPORT	t0–t6	0.029	0.217	0.039	−0.030	−0.027	−0.166	−0.077	−0.126	−.060	−0.148
	t0–t12	0.124	0.290 *	−0.077	−0.053	−0.101	−0.120	−0.184	−0.103	−0.205	−0.131
**FBIS−II**											
Total presence of burden	t0–t6	0.095	0.126	−0.057	−0.203	−0.163	−0.164	−0.286 *	−0.293 *	−0.254 *	−0.251 *
	t0–t12	0.000	0.011	−0.023	−0.134	−0.025	−0.077	−0.167	−0.197	−0.112	−0.185
Sum frequency of burden	t0–t6	0.098	0.130	−0.101	−0.142	−0.186	−0.133	−0.236 *	−0.202	−0.179	−0.114
	t0–t12	0.047	0.117	−0.113	−0.138	−0.096	−0.129	−0.191	−0.191	−0.095	−0.138
Average frequency of burden	t0–t6	0.116	0.013	−0.111	−0.001	−0.138	−0.048	−0.090	−0.037	−0.092	0.043
	t0–t12	0.064	0.063	−0.165	−0.093	−0.109	−0.135	−0.144	−0.120	−0.084	−0.090
Sum concern about burden	t0–t6	0.030	0.256 *	−0.004	−0.320 *	−0.116	−0.376 **	−0.181	−0.402 **	−0.137	−0.360 **
	t0–t12	0.090	0.066	−0.008	−0.192	−0.083	−0.215	−0.197	−0.260 *	−0.100	−0.210
Average concern about burden	t0–t6	−0.029	0.023	0.022	−0.097	0.023	−0.174	0.100	−0.041	0.049	−0.030
	t0–t12	0.013	−0.237	0.040	0.109	0.055	0.043	0.013	0.125	0.008	0.151
Total time	t0–t6	0.033	0.221	−0.193	−0.241	−0.337 **	−0.348 **	−0.369 **	−0.273 *	−0.264 *	−0.178
	t0–t12	0.115	0.251	−0.190	−0.224	−0.283 *	−0.300 *	−0.352 **	−0.269 *	−0.183	−0.174
TOTAL FBIS-II	t0–t6	0.022	0.388 **	−0.100	−0.352 **	−0.171	−0.362 **	−0.311 **	−0.448 **	−0.226	−0.428 **
	t0–t12	0.084	0.225	−0.044	−0.227	−0.039	−0.172	−0.263 *	−0.356 **	−0.142	−0.357 **

* *p* < 0.05, ** *p* < 0.01.

**Table 4 ijerph-19-13935-t004:** Pearson correlations between service users’ and relatives’ participation and scores at all time points.

	Service Users’ Participation			Relatives’ Participation		
	t0	t6	t12	t0	t6	t12
**STORI**						
Moratorium	−0.128	−0.283 **	−0.234 *	−0.153	0.008	−0.136
Awareness	0.124	0.217 *	0.159	0.129	0.136	0.085
Preparation	0.114	0.246 *	0.296 *	0.146	0.189	0.213
Rebuilding	0.082	0.027	0.167	0.131	0.071	0.173
Growth	0.047	0.146	0.241 *	0.200 *	0.076	0.230 *
**Duke**						
Affective Social Support	0.000	−0.006	−0.083	−0.043	0.084	0.039
Confident Social Support	−0.089	−0.101	−0.225 *	−0.173	0.025	−0.047
GLOBAL SOCIAL SUPPORT	−0.056	−0.064	−0.172	−0.13	0.055	−0.011
**FBIS−II**						
Total presence of burden	−0.183	−0.130	−0.033	0.031	0.087	0.21
Sum frequency of burden	−0.231 *	−0.210	−0.065	0.050	0.073	0.156
Average frequency of burden	−0.223 *	−0.254 *	−0.089	0.048	0.057	0.135
Sum concern about burden	−0.088	−0.225 *	−0.072	0.106	0.015	0.161
Average concern about burden	0.119	−0.173	−0.066	0.141	0.004	−0.062
Total time	−0.174	−0.244 *	−0.074	0.035	−0.083	0.037
TOTAL FBIS-II	−0.116	−0.268 *	−0.088	0.096	0.012	0.094

* *p* < 0.05, ** *p* < 0.01.

**Table 5 ijerph-19-13935-t005:** Mixed models’ type III effects of time and participation.

	Service Users				Relatives			
	Time		Participation		Time		Participation	
	F	*p*	F	*p*	F	*p*	F	*p*
**STORI**								
Moratorium	14.531	<0.001 ***	4.468	0.037 *	16.187	<0.001 **	0.848	0.359
Awareness	1.306	0.256	6.524	0.12 *	1.820	0.181	2.680	0.105
Preparation	2.634	0.108	8.524	0.004 **	3.336	0.071	5.666	0.019 *
Rebuilding	5.650	0.020	1.961	0.164	5.957	0.017 *	2.549	0.113
Growth	3.831	0.054	2.375	0.126	4.024	0.048 *	4.745	0.032 *
**Duke**								
Affective Social Support	1.119	0.293	0.151	0.699	1.072	0.303	0.005	0.944
Confident Social Support	1.164	0.284	3.058	0.083	1.152	0.286	1.478	0.227
GLOBAL SOCIAL SUPPORT	1.298	0.258	1.386	0.258	1.280	0.261	0.539	0.465
**FBIS−II**								
Total presence of burden	20.709	<0.0001 ***	2.971	0.088	22.315	<0.0001 ***	0.318	0.574
Sum frequency of burden	31.698	<0.0001 ***	4.132	0.044 *	34.428	<0.0001 ***	0.480	0.490
Average frequency of burden	65.893	<0.0001 ***	5.382	0.022 *	70.003	<0.0001 ***	0.576	0.453
Sum concern about burden	19.179	<0.0001 ***	1.968	0.163	22.012	<0.0001 ***	1.309	0.255
Average concern about burden	15.854	<0.001 ***	0.091	0.764	16.538	<0.0001 ***	1.111	0.294
Total time	14.748	<0.001 ***	3.859	0.052	15.394	<0.001 ***	1.326	0.251
TOTAL FBIS-II	12.153	<0.001 ***	3.185	0.076	7.500	0.007 **	0.619	0.432

* *p* < 0.05, ** *p* < 0.01, *** *p* < 0.001.

## Data Availability

A database and calculations syntax can be downloaded as Appendix A.

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
