# Peer review of "Interaction between the Participation in and the Impact on Mental Health Service Users and Their Relatives of a Multicomponent Empowerment-Based Psychosocial Intervention"

_ijerph, 2022, doi:10.3390/ijerph192113935_

Round 1

Reviewer 1 Report

Thank you for the opportunity to review the manuscript "Interaction between participation and impact of a multicomponent empowerment-based psychosocial mental health intervention on service users and their relatives".

The work is  important and have a contribution to the adressed field. Despite the interesting issue and results I would like to make some comments that should be addressed:

 Abstract: (1) What is RCT? I guiss it is an abbreviation, but of what? the authors did not give information about RCT. (2) I think that the authors should add more information about the sample, such as sample size and subject ages and maybe any other relative informations in the Abstract.

 Introduction:  the introduction is clear and well written. However, the authors should clarify the quistions and the hypothoses of the research.

Methods: 1) As a suggestion, I recommend to perform a network analysis that gives deeper information about the links between the variables, and what is the most central parts.

Results: 1) I recommend to give more data about the General lineal models section. 2) The authors may add one figure at least to illustrate the results.

Author Response

Dear Reviewer,

We would like to resubmit the manuscript retitled “Interaction between participation and impact of a multicomponent empowerment-based psychosocial intervention on mental health service users and their relatives”. We have tried to address all the issues raised and have accommodated and incorporated all the necessary revisions. Below you can find all the points raised and our respective annotation to the modification.

First, we have uploaded a clean version of the manuscript including tables and figures, followed by a tracked changes version of the modified documents in the Editorial Manager application.

We look forward to your comments.

Sincerely,

The authors

Reviewer 2 Report

Brief summary:

This is an interesting study that aims to establish the mutual interaction effects of the participation and outcomes of services users and their relatives benefiting from the Activa’t per la Salut Mental multicomponent interventions. It is a correlational longitudinal study based on the Activa’t pilot implementation RCT study. The interventions are: (1) non-therapeutic counselling service, (2) a psychoeducation program, (3) an empowerment training program and (4) self-managed peer support groups. The authors highlight in this study that the level of participation of both service users and relatives correlated with baseline characteristics and scores, as well as the scores during the trial. This article is pertinent to the field as it provides an insight of the wellbeing of mental health service users and their relatives. Please see below my comments.

COMMENTS:

Introduction:

The introduction is well written. Considering the importance of the service user’s relatives in your study, I would suggest adding a few lines in the second paragraph to highlight the importance of relatives in the recovery process of people suffering from mental illness. Relatives are important and there is an interesting body of literature on the topic.

When you refer to the Recovery program, I would suggest using Italic to avoid confusion with the term recovery.

While implicit, I would suggest adding your hypothesis after line 74 in lights of the phenomenon previously observed in the pilot implementation of the intervention.

Minor comment: line 65: I suggest removing the word ‘’packs’’ considering it is a succession of four interventions (not four packages of interventions).

Materials and Methods:

The methodology is well described and easy to follow. In the participants section, the diagnosis are they clinical diagnosis? If so, is it per DSM or ICD? If it is per DSM, I would suggest using the proper nomenclature (ex.: delusional should be delusion disorder ; schizotypal is not an entity in the DSM).

Line 109: what is a participation compromise?

In the sample size section, the authors mention that the MDE is 240 family units and the current sample size is 222 (line 119). Why (how) was 222 determined sufficient? I would suggest clarifying this gap.

Results:

Table 1 could benefit from adding the proportion of diagnosis that met the inclusion criteria.
Otherwise, the results section is clear and concise.

Discussion:

I would suggests rephrasing the beginning of line 273 (Before entering into details on these results) as this is not in a neutral scientific tone. The discussion is very short in its current form. It would benefit from further discussions with the literature on the subject.

Lines 293-296: I am unsure what the authors meant by this sentence. I would suggest rephrasing.

Line 298: severe mental disorder was not defined previously in your manuscript. The inclusion criteria include vague diagnosis such as schizotypal which does not directly imply a severe mental disorder.

Overall, this paper is very interesting and provides an insight on how Activa't contribute to ease the relatives' burden and contribute to users' recovery.

Minor comment: the Fundings section seems to be filled with the template information.

Author Response

Dear Reviewer,

We would like to resubmit the manuscript retitled ‘Interaction between participation and impact of a multicomponent empowerment-based psychosocial intervention on mental health service users and their relatives’. We have tried to address all the issues raised and have accommodated and incorporated all the necessary revisions. Below you can find all the points raised and our respective annotation to the modification.

First, we have uploaded a clean version of the manuscript including tables and figures, followed by a tracked changes version of the modified documents in the Editorial Manager application.

We look forward to your comments.

Sincerely,

The authors

Reviewer 3 Report

Interesting and well prepared paper. Nevertheless some minor issues need to be adressed.

Abstract in the detailed way describes the methodology, however is a little bit vague in the results part. I would suggest more specific results presentation.

The tables' graphic form shoud be adjusted - the statistical significance, depending on ther table, is indicated by bold, asterisk or both - it is confusing.

One of the findings is difference of educational level between high and low participating group. This should be discussed. I believe that practical implications of this finding may be explored. Possibly the form or content of the interventional sessions can be adjusted to the educational level of participants?

In general, the discussion is rather brief. I would like to know what practical implications, further hypotheses etc. could be drawned from the presented results.

Author Response

(The authors gave the same response as above.)
